# Identifying healthcare needs with patient experience reviews using ChatGPT

**Jiaxuan Li[1], Yunchu Yang[1], Rong Chen[2], Dashun Zheng[1], Patrick Cheong-Iao Pang**[iD][1]*,
**Chi Kin Lam[1], Dennis Wong[1,3], Yapeng Wang**[iD][1]

**1** Faculty of Applied Sciences, Macao Polytechnic University, Macao, China, **2** Department of Rehabilitation Medicine, The First Affiliated Hospital, Sun Yat-Sen University, Guangzhou, China, **3** State University of New York, Songdo, Korea

* mail@patrickpang.net

## Abstract

### Background

Valuable findings can be obtained through data mining in patients' online reviews. Also identifying healthcare needs from the patient's perspective can more accurately improve the quality of care and the experience of the visit. Thereby avoiding unnecessary waste of health care resources. The large language model (LLM) can be a promising tool due to research that demonstrates its outstanding performance and potential in directions such as data mining, healthcare management, and more.

### Objective

We aim to propose a methodology to address this problem, specifically, the recent breakthrough of LLM can be leveraged for effectively understanding healthcare needs from patient experience reviews.

### Methods

We used 504,198 reviews collected from a large online medical platform, haodf.com. We used the reviews to create Aspect Based Sentiment Analysis (ABSA) templates, which categorized patient reviews into three categories, reflecting the areas of concern of patients. With the introduction of thought chains, we embedded ABSA templates into the prompts for ChatGPT, which was then used to identify patient needs.

### Results

Our method has a weighted total precision of 0.944, which was outstanding compared to the direct narrative tasks in ChatGPT-4o, which have a weighted total precision of 0.890. Weighted total recall and F1 scores also reached 0.884 and 0.912 respectively, surpassing the 0.802 and 0.843 scores for "direct narratives in ChatGPT." Finally, the accuracy of the three sampling methods was 91.8%, 91.7%, and 91.2%, with an average accuracy of over 91.5%.

**Data availability statement:** The dataset used for the experiments in this paper is archived at Zenodo (DOI: 10.5281/zenodo.13995225).

**Funding:** The support received during this research was funded by the Macao Science and Technology Development Fund (funding IDs: 0048/2021/APD; 0088/2023/ITP2). The publication costs were supported by Macao Polytechnic University (submission approval code: fca.8933.d005.1).

**Competing interests:** The authors have declared that no competing interests exist.

**Abbreviations:** LLM: Large Language Model; NLP: Natural Language Processing; CoT: Chain of Thought; ABSA: Aspect based sentiment analysis.

## Conclusions

Combining ChatGPT with ABSA templates can achieve satisfactory results in analyzing patient reviews. As our work applies to other LLMs, we shed light on understanding the demands of patients and health consumers with novel models, which can contribute to the agenda of enhancing patient experience and better healthcare resource allocations effectively.

## Introduction

The challenge of efficiently assessing and distributing medical resources is critical for governments and healthcare organizations [1]. Overutilization of medical resources is a worldwide issue that requires immediate attention [2], and traditional assessment tools often fail to capture the actual demands, leading to suboptimal resource allocation [3]. Identifying patients' needs from their feedback can help justify healthcare resource allocation and improve service quality [4,5]. Therefore, incorporating patient feedback is essential because it provides a realistic evaluation of healthcare services, which is often overlooked.

Recent research, including the use of deep learning and data mining approaches, has begun to analyze patient reviews to improve assessment methods [6–8], but these techniques still struggle to pinpoint specific areas of patient dissatisfaction within individual reviews [9,10]. This results in the patient experience not being adequately met in a timely manner, while it is very important in the modern healthcare system, because have research has pointed out that the patient's visit experience contributes to the quality of healthcare services [11]. At the same time, researchers have demonstrated that the doctor's visit skills and efficiency can reflect the doctor's professional laxity is high, the probability of medical errors rises dramatically [12]. It has also been noted that optimization of administrative issues can significantly improve the patient experience [13].

The introduction of ChatGPT has brought LLMs into the public eye, and LLMs are being adopted in various fields, including medicine [14–16]. Among the research in LLM within the medical field include the directions of health counseling [17], health education [17], medication management [18], and online consultation with e-health [19]. On the other hand, some researchers mentioned that LLM has great potential in the direction of healthcare [20], which includes administration and management in healthcare [21], while the analysis of patient satisfaction is one of the potential innovative applications [21]. In another study, a research team noted that the best way to deal with a patient's problem depends on the patient's situation and feelings [22]. Finally, it was also mentioned that Natural Language Processing (NLP) technology is expected to prove to be an integral part of addressing healthcare inequalities [22].

Similarly, some researchers mentioned that LLM in patient-driven healthcare management is expected to enhance the patient experience [23]. And illustrates that LLM has been shown great potential in patient-centered healthcare [24]. Specifically, LLM can streamline administrative processes as well as improve efficiency, which can reduce the administrative burden on physicians [23]. Nevertheless, scholars have noted that LLMs can exhibit erratic and unstable behavior during their operation [25]. In addition, as the size of the resultant data output using LLM becomes larger, it becomes difficult to assess the accuracy of all the results [26]. Consequently, it is necessary to construct the evaluation methods for the large-scale unlabeled data results.

Therefore, we proposed the ABSA templates, which can help us categorize patient reviews into "patient experience," "physician skills and efficiency," and "infrastructure and administration." Additionally, we embedded ABSA templates into ChatGPT-4o and successfully accomplished the task of categorizing patient reviews into three categories. We comprehensively evaluated the feasibility and reliability of this method in two different ways: "model performance on a test set" and "manual evaluation results." The results show that this method

maintains excellent accuracy, although large models experience performance degradation over long periods.

## Methods

Our approach to using ABSA templates with ChatGPT relies on prompt engineering. Currently, prompt engineering plays a key role in using LLM, and an excellent prompt design can better control LLMs for constraining the outputs from being in line with the requirements and avoid problems such as hallucinations, and uncontrollable outputs, which generative AI models commonly produce. More specifically, our approach described the patient review data to be processed and the tasks that need to be performed by ChatGPT, as well as the format of outputs and other instructions with prompting to facilitate the standardized extraction of healthcare needs. In this section, we elaborate on the data used for analyses, the details of ABSA, and the evaluation methods.

### Hyperparameterization

There are several adjustable hyperparameters within ChatGPT, such as temperature, top_p, frequency_penalty, and presence_penalty. We did not change these hyperparameters to ensure the generalizability of the experiments to the most common experimental environments in which we evaluated the performance of our model. Accordingly, we have not altered the default settings, where temperature was set to 1.0, top_p was set to 1.0, and frequency_penalty and presence_penalty were set to 0.

### Data

The data used in this project was collected from haodf.com. It is considered the most prominent online medical platform in mainland China, with more than 6.1 million patients uploading their visit experience [27]. Studies have indicated that ChatGPT's ability of processing Chinese data directly [28], which does not require additional translations. For illustration purposes, we translated a sample patient review into English, as shown in Fig 1. The review includes the name of the condition, treatment type, treatment outcome, the date of consultation, location, the name of the hospital, the name of the physician, and a patient review. The language of our dataset is Chinese, and our experiments in LLM are conducted using the Chinese dataset.

These patient reviews are verified by the platform to ensure they are commented on by real patients with genuine experience of medical consultations. In this study, we crawled 552,764 unique patient reviews from six regions of China in haodf.com. The data were crawled from November 2022 to June 2023. We did not manually filter or remove any data. The collected patient reviews ranged from October 2003 to June 2023. 504,198 reviews remained after the preliminary data cleaning including the removal of incomplete data and duplicates. Individually identifiable information about participants was consistently unavailable to the authors during or after data collection, and all patient information was made private by the platform (haodf. com).

### Ethics approval

This study used anonymized data that was publicly available on the Internet. Therefore, Human Ethics and Consent to Participate declarations: not applicable.

### ABSA templates

Current methods can only perform sentiment classifications for reviews as a whole. In a lengthy review (as illustrated in Fig 1), there are positive and negative sentiments towards different aspects of the healthcare process. To address this problem, we designed ABSA based

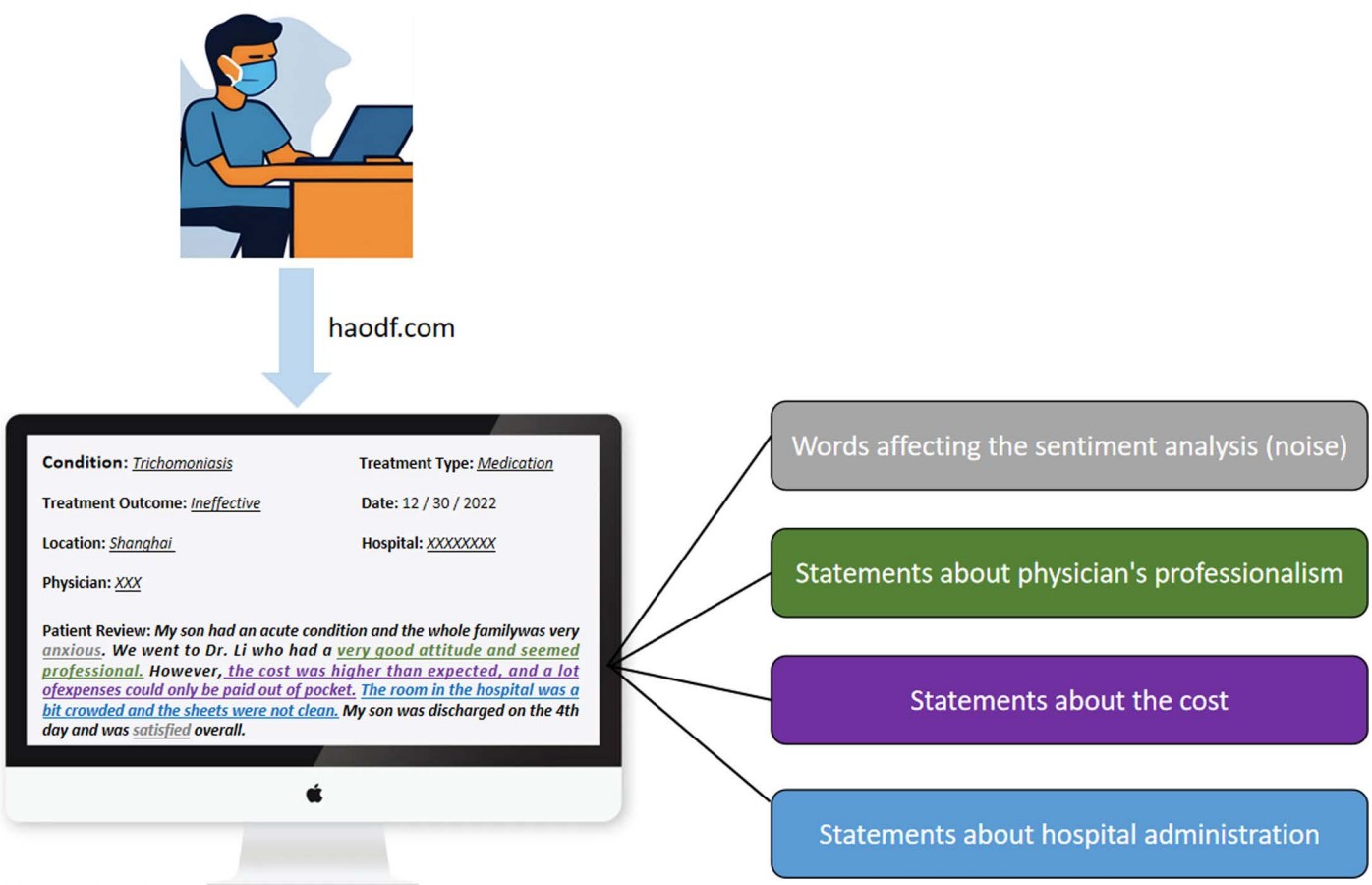

**Fig 1. An example of patient reviews with a mix of healthcare needs and sentiments.**

on existing literature and data. Other research divided patients' review content into several dimensions for analyses [29,30]. A researcher listed the dimensions of tools used for appraising patient satisfaction in healthcare since 1974. There are 12 versions, and we selected the most recent version for revision [31]. Also, a study [31] noted the validity of the tool for comprehensive analysis of patient comments. We use the latest version of the tool as a basis [31], and it is modified with the experience of daily operations of the hospital in where author RC works. We also used the characterization of the research data from this project as a basis for modification.

As shown in Fig 2, we changed "clinical atmosphere" to "patient experience" in the original tool [31]. Since most hospitals in China do not have general practitioners for orientation, more consultation steps are needed before the actual clinical visit, and the abovementioned tool does not reflect the complete feelings and experiences of the patients. In this case, we changed it to the "patient experience" includes patients' perspectives about the physician and the overall experience of their hospital visits. This type of feedback can contribute to the positive change and quality improvement of healthcare delivery in response to patient needs [11,32]. Compared to the previous one, the "patient experience" is more oriented to all the patients' feelings rather than being limited to the clinical experience.

Secondly, we analyzed the data with a data-driven mindset and found that "treatment process" and "care outcome" were often combined in the same sentence in patient reviews. At the same time, "skills" and "efficiency" also appeared together. Therefore, we grouped all these

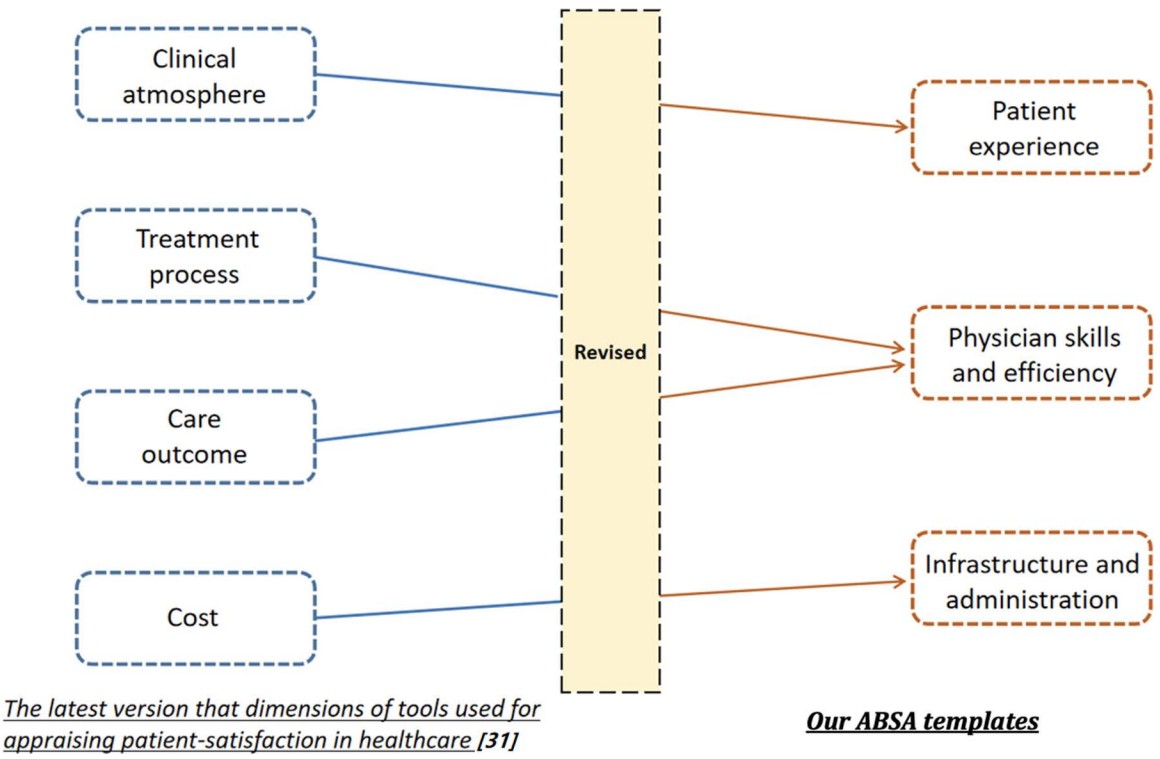

**Fig 2. Schematic diagram of the revision process of ABSA templates.**

elements into one category, "physician skills and efficiency," to better incorporate the data types in the patient reviews.

On the other hand, "physician skills and efficiency" includes the physician's treatment plan for the disease, the relief of the patient's disease, and the physician's treatment efficiency. This category also includes patients' evaluations of the physician's professionalism, and the negativity in this category can imply adverse consequences for patients and society, including increased healthcare costs and even increased mortality rates [17].

Finally, combining data content with clinical experience, we believe categorizing "cost" alone is too homogenous. For example, we found that there were grievances in the data about hygiene in hospitals and difficulties in parking, which would not have been categorized using the original tool. We, therefore, broadened the scope by changing it to "infrastructure and administration," which includes the conditions of medical facilities, environment, hygiene, registration, and other administrative issues of the hospital, which significantly affect patient experience and the length of stay [19,33].

In summary, our ABSA templates consist of three categories, the first being "patient experience," the second "physician skills and efficiency," and the third "infrastructure and administration." As explained above, these three categories are all clinically significant [32–33]. Examples of these categories are shown in Table 1. Positive and negative data of various categories include but are not limited to, the range shown in the examples.

## Chain of thought (CoT)

Differentiating the sentiment of review subjects in reviews is a process that requires logical reasoning; therefore, we used the CoT technique in prompting to make LLMs think step-by-step and not skip or omit steps. With CoT, an LLM is guided to analyze reviews to see if the

**Table 1. Positive and negative examples of ABSA templates.**

|  | Positive examples | Negative examples |
|---|---|---|
| Patient experience | *The physician was very friendly, and the overall experience of the visit was quite satisfying.* | *No consideration for the patient whatsoever, and it was a truly dreadful experience.* |
| Physician skills and efficiency | *The eye drops prescribed by the physician, azelastine hydrochloride, quickly alleviated the symptoms, and they haven't returned since.* | *Prescribing medication without discrimination, experiencing constant side effects, vomiting persistently, and nothing has proven effective so far!* |
| Infrastructure and administration | *The hospital is exceptionally clean, and it boasts a top-notch infrastructure equipped with state-of-the-art instruments!* | *It's tough to sign up, and there are these random fees!* |

evaluation of each subject occurs and generate the needed answer at the end. Compared to the output without CoT, the accuracy is improved, and it is convenient to check the reasoning logic of the model. Finally, we used the zero-shot technique and added an example in the prompt for the model to learn to control the model's output.

After the LLM receives our task, the CoT forces the LLM to complete the task step by step. Our CoT prompting method works specifically, as shown in Fig 3. First, when ChatGPT receives a comment from a patient. The first step of the model is to extract all the sentences related to "patient experience" from this comment, and then the second step is to analyze its sentiment. After completing the sentiment analysis for the first category, the model will extract all the sentences about "physician skills and efficiency" from the patient's comment in the third step and then perform the sentiment analysis. Finally, the model will extract the sentences about "infrastructure and administration" for sentiment analysis and then output the results of all the sentiment analyses.

Our prompt is written in English, following other work [34,35] suggesting the use of English for better performance with ChatGPT. Another study also states that CoT can archive better performance in English [36]. The content of the prompt is shown below.

> *"You are a review data processing expert and your task is to perform a sentiment analysis on a user's visit review. Segment the comments into attitudes toward three aspects: 1. Patient experience short as PE, patients' perspectives about the physician and the overall experience of their hospital visits. 2. Physician skills and efficiency short as PSE, mainly include the physician's treatment plan for the disease, the relief of the patient's disease, and the physician's treatment efficiency. 3. Infrastructure and administration short as ID, include the conditions of medical facilities, cost, environment, hygiene, registration, and other administrative issues of the hospital, Mark each type of sentiment using Positive, Negative, and None. Give your reasoning and output the final result at the end in the following format: {PE: positive, PSE: positive, ID: none}. Please conduct a sentiment analysis based on the following comments {content} Let's think step by step."*

## Proposed architecture

Fig 4 shows the proposed architecture for identifying healthcare needs from patient reviews. In this architecture, ABSA templates are used with CoT to generate a prompt question, fed into an LLM with the patient review content. To consider the practicality and cost, we chose the ChatGPT-4o as the LLM for the experiment since it is the most cost-effective one. In the output, the model is expected to give the results of the sentiments for three different categories. For instance, the sample review shown in Fig 4 returns positive sentiment for the "patient experience" category but negative for "physician skills and efficiency" and "infrastructure and administration" categories.

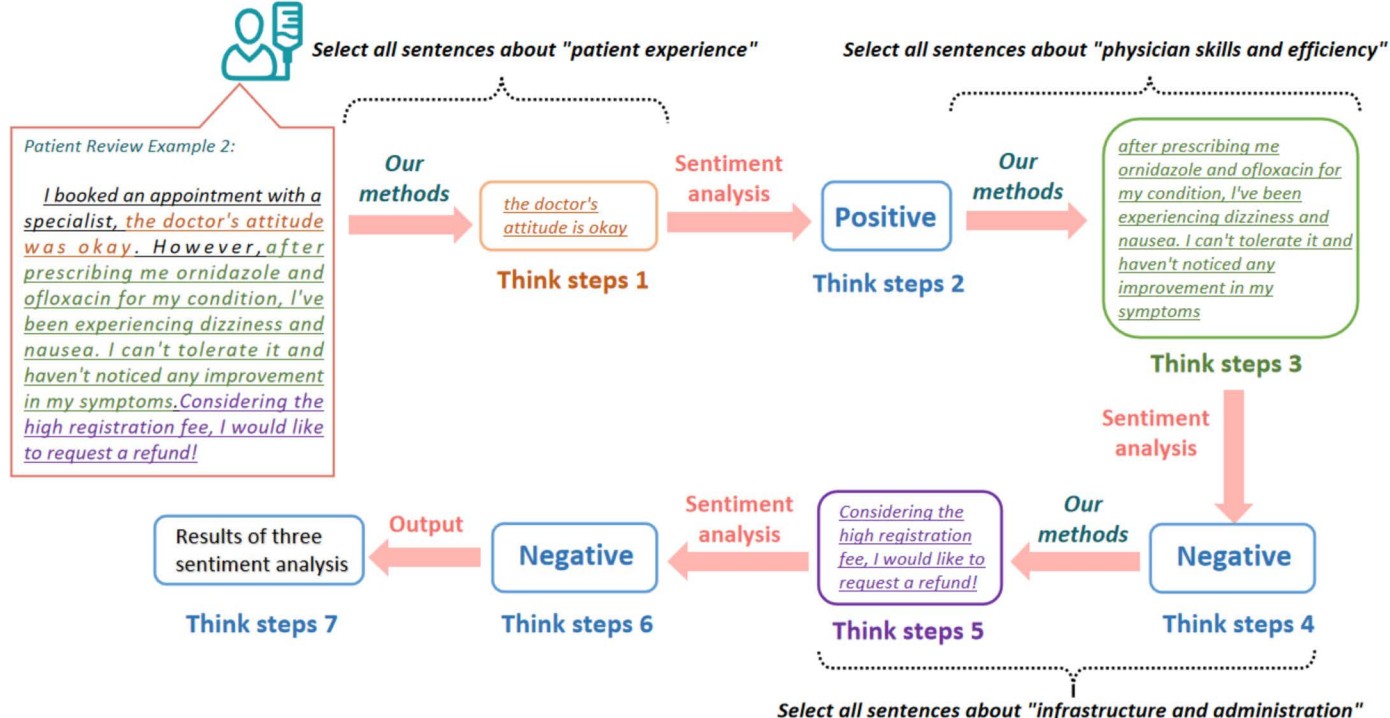

**Fig 3. Our methods of thinking process diagram.**

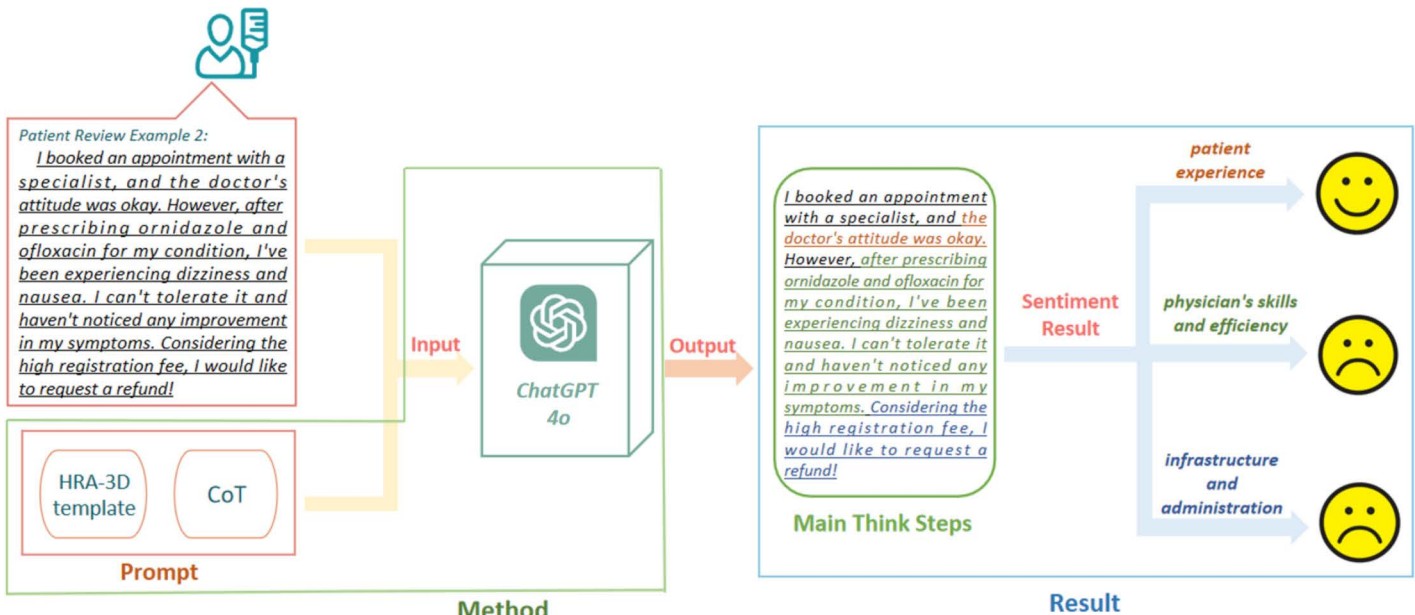

**Fig 4. The proposed architecture with ABSA and ChatGPT for analyzing patient reviews.**

### Evaluation method

In this work, we employ two approaches to assess the accuracy of our method. On the one hand, we tested the performance of our method with a manually labeled test set and computed metrics such as precision, recall, and F1 scores, respectively. On the other hand, evaluating the results by manual random sampling can increase the interpretability of the results and make them more comprehensive and credible.

Also, the idea of randomization ensures the representativeness of our results as well as avoids potential biases. Therefore, we used a double criterion to evaluate the outputs comprehensively. Finally, we evaluated the cost and runtime to assess its applicability in real-world applications.

**Model performance.** We manually annotated the patient review data with three categories defined in ABSA, and each category in the same review was labeled as either "positive" or "negative." If the review did not mention a certain category, it was labeled "none." The labeling was performed by multiple authors with healthcare backgrounds. We used random sampling to randomly select 2000 comments out of all the data as a test set. We labeled these 2000 reviews and used them as a test set to evaluate precision, recall, and F1 scores. Finally, we also collected results in three categories in ABSA. Additionally, to verify that our approach improves on the direct narrative tasks in ChatGPT (i.e., the case without ABSA and CoT), we compared the precision, recall, and F1 Scores to the direct narrating tasks to ChatGPT.

The F1 score, which combines precision and recall, is well-tried in the case of multiclassification and uneven classification and is a metric often used in sentiment classification tasks. Therefore, we introduce the F1 score as a metric for evaluating our sentiment classification method. The formula calculates the general F1 score:

$$\text{Precision} = \frac{\text{TP}}{\text{TP} + \text{FP}} \tag{1}$$

$$\text{Recall} = \frac{\text{TP}}{\text{TP} + \text{FN}} \tag{2}$$

$$\text{F1} = \frac{2 * \text{Precision} * \text{Recall}}{\text{Precision} + \text{Recall}} \tag{3}$$

Where TP refers to True Positive, representing the number of samples correctly predicted as positive cases, FN is False Negative, representing the number of samples the model fails to predict as positive cases. FP refers to False Positive, representing the number of samples the model incorrectly predicts as positive cases. To avoid sample imbalance affecting the correctness of our conclusions, we chose to calculate the model's score using the modified weight F1 score with the formula:

$$\text{F1}_{weight} = \sum_{i-1}^{n} w^i F^i \tag{4}$$

Where $w^i$ represents the weighting of the different classes and $F^i$ is the F1 score value for each class individually. We used the default function for weight F1 score in our calculations from sklearn, a scientific computing library for Python, whose calculations use the percentage of a class in the total sample as the weight for the calculations.

**Reliability and stability evaluation.** Although we tested the model's accuracy using a manual-labeled test set, several studies have pointed to instability and unreliability in LLMs

[37,38]. In this case, we perform a mixture of three random sampling methods to assess the reliability and stability of the LLM model in our task. We used all three random sampling methods because several studies have pointed out the merits of each of them [39–41]. The details of the three specific sampling modes are shown in Table 2.

Mode 1 is simple random sampling [39,41]. We randomly selected 2,000 of the 504,198 datasets for evaluation. Mode 2 is incremental random sampling [42], where we take 200 reviews for the first time, followed by another 400 reviews, 600 reviews, and 800 reviews in each round. Mode 3 is equidistant sampling [40], where 500 reviews are randomly selected in each turn with a total of 4 times.

## Money and time costs

In addition to the accuracy of LLM, we also pay attention to the possibility that this approach can be used practically in healthcare providers and other stakeholders. In this regard, we evaluate the monetary cost as well as the time cost of processing data. Since the billing unit in ChatGPT is not word-based but token-based, we additionally count the tokens required for the operations.

Specifically, we calculated the average text length, token length, the token length of prompts, the total token length, the model processing time, the fees for ChatGPT payments, and the average fee for each patient review.

## Results

In this section, we describe the model performance and manual randomization accuracy to demonstrate the degree of reliability of LLMs for such healthcare tasks. Subsequently, we will characterize the negative rates of sentiment analysis for the three categories in ABSA to show the extent of healthcare needs in the different categories. Finally, we will show the statistically obtained monetary cost and the time cost to demonstrate the degree of likelihood of landing an LLM in healthcare providers.

### Model performance

As shown in Table 3, the weighted total performance of our method outperforms "direct narrative tasks in ChatGPT" in terms of precision, recall, and F1 score. In the F1 score, we reach 0.912, which is 6.9% higher than it. Secondly, we outperformed "direct narrative tasks in ChatGPT" in precision and recall by 5.4% and 8.2% respectively. For this task, a higher F1 score, Precision, and Recall indicate that the model is more capable. The F1 score is a more comprehensive performance indicator, and Precision is closer to the real accuracy of this task. To avoid sample imbalance and to improve readability, the F1 score presented in the Result section are weighted F1 score.

Secondly, our method achieves an impressive 0.957 (F1 score), 0.964 (precision), and 0.952 (recall) in the "patient experience" category. This performance is 0.084, 0.053, and

**Table 2. Three modes of sampling methods.**

| Mode | Sampling | | | |
|---|---|---|---|---|
| | 1st | 2nd | 3rd | 4th |
| Mode 1 | 2000 | — | | |
| Mode 2 | 200 | 400 | 600 | 800 |
| Mode 3 | 500 | 500 | 500 | 500 |

**Table 3. Comparison table of model performance results.**

|  |  | F1 score | Precision | Recall |
|---|---|---|---|---|
|  | Patient experience | 0.957 | 0.964 | 0.952 |
| Ours | Physician skills and efficiency | 0.879 | 0.918 | 0.845 |
|  | Infrastructure and administration | 0.899 | 0.951 | 0.854 |
|  | **Weighted total** | **0.912** | **0.944** | **0.884** |
| Direct narrative tasks in ChatGPT | Patient experience | 0.873 | 0.911 | 0.838 |
|  | Physician skills and efficiency | 0.840 | 0.867 | 0.815 |
|  | Infrastructure and administration | 0.816 | 0.892 | 0.752 |
|  | **Weighted total** | **0.843** | **0.890** | 0.802 |

0.114 higher than "direct narrative tasks in ChatGPT," respectively. The performance of this category is also the best among the three categories in ABSA. Third, in the "physician skills and efficiency" category, our F1 score, precision, and recall are 0.879, 0.918, and 0.845, respectively, higher than those of "direct narrative tasks in ChatGPT." are 0.840, 0.867, and 0.815, respectively.

Finally, in the "infrastructure and administration" category, the F1 score decreases with low recall. Although the recall of our method is only 0.854, it is still 13.6% higher than that of "direct narrative tasks in ChatGPT." In terms of F1 score and precision, we also outperform it by 8.1% and 10.2%. Overall, our method improved all metrics across all classifications, and some metrics accomplished huge improvements. Meanwhile, the 0.944 total precision is an outstanding performance value.

## Time and money costs

We measured the time and costs used under the model based on ChatGPT-4o-latest. The average text length and token length of reviews are 52 and 69, and the average total token length after appending our prompt is 407. The average cost of processing per 1000 review was USD 1.9. Additionally, we counted the processing time per 1000 reviews based on our duration of study. Our study ran for 21 days, in which the average processing time for every 1000 reviews was 55 (range 51-59) minutes.

## Model reliability and stability

As shown in Table 4, we used three ways to sample the total data, and the results show that the total average accuracy remained above 91% regardless of the sampling method, with Mode 1 having the highest total average accuracy of 91.8%, followed by Mode 2 with 91.7% and Mode 3 with 91.2%. The "patient experience" category had the highest accuracy of all figures.

## Discussion

### Principal results

**Main contributions.** This study makes two significant contributions. Firstly, we integrate ABSA templates into ChatGPT, demonstrating their effectiveness through various evaluation methods and establishing their credibility in the field. Secondly, we provide a framework with feasible potential clinical applications, within a specific domain and the introduction of two evaluation methods for LLMs: test sets for performance evaluation and multifaceted result sampling. These methods ensure both initial performance validation and long-term model stability. At the same time, we provide many categories of applicability, such as the use of big data to discover valuable insights in healthcare information from different patient perspectives,

**Table 4. Reliability and stability on accuracy.**

| Mode | Sampling | | | | Average accuracy of the entire sample |
|---|---|---|---|---|---|
| | **1st** | **2nd** | **3rd** | **4th** | |
| Mode 1 | *2000* | | | | **91.8%** |
| Sample Size (N) | | | | | |
| *Accuracy:* | | | | | |
| Patient experience | 94.70% | | | | |
| Physician skills and efficiency | 91.10% | | | | |
| Infrastructure and administration | 89.50% | | | | |
| Mode 2 | | | | | **91.7%** |
| Sample Size (N) | *200* | *400* | *600* | *800* | |
| *Accuracy:* | | | | | |
| Patient experience | 93.50% | 92.00% | 92.50% | 91.10% | |
| Physician skills and efficiency | 95.50% | 91.30% | 94.80% | 91.60% | |
| Infrastructure and administration | 91.00% | 91.80% | 87.20% | 88.60% | |
| Mode 3 | | | | | **91.2%** |
| Sample Size (N) | *500* | *500* | *500* | *500* | |
| *Accuracy:* | | | | | |
| Patient experience | 91.40% | 92.20% | 91.60% | 88.60% | |
| Physician skills and efficiency | 92.60% | 95.20% | 93.20% | 89.20% | |
| Infrastructure and administration | 88.60% | 91.00% | 88.80% | 92.20% | |

such as the patient experience, physician skills and efficiency, as well as infrastructure and administration. Finally, this framework also contributes to potential applications of using LLMs for government policy reliability assessment, policy refinement, and macro-assessment of the distribution of healthcare resources among regions, hospitals, and departments.

**Model accuracy and feasibility.** We labeled a patient review dataset and evaluated the model performance regarding accuracy, reliability, and stability. The results show that our model's performance is 0.912 (F1 score), 0.944 (precision), and 0.884 (recall), much higher than describing the same tasks directly submitted to ChatGPT, demonstrating the effectiveness of our combined architecture of ABSA templates and CoT. Additionally, in stability and reliability testing with three modes of random sampling, the accuracy values reached 91.8%, 91.7%, and 91.2%, respectively.

As shown in Table 3, our precisions in each task are very inspiring, especially with a total precision of 0.944. Precision is the percentage of samples correctly predicted by the model out of the total number of positively predicted samples. Therefore, it is most representative of our ABSA sentiment analysis task. This is because it represents the avoidance of categorizing one sentiment (e.g., negative) as another (e.g., positive). Also, this is very close to the accuracy of our manual randomized assessment, which validates that precision is the most appropriate for assessing the confidence of our task too.

Although our recall for "infrastructure and administration" has increased by 10.2% compared to the recall for "direct narrative tasks in ChatGPT." But it's still only 0.854. One of the reasons for this is that most patient comments are still only about "patient experience" and "physician skills and efficiency." The relative scarcity of "infrastructure and administration" in the datasets has resulted in the model being unable to capture many negative comments, hence the low recall rate. In summary, after observing examples of LLM's misclassification of patient comments, we have found that LLM still fails to find deeper emotions when dealing with some complex emotions. For example, some angry patients hypocritically "praise" the hospital, but the LLM cannot accurately categorize such emotions.

To increase the potential of using our approach practically for analyzing patient reviews, we used ChatGPT-4o, which is the most popular LLM in the ChatGPT series. Using our architecture shows a high accuracy rate of over 91% with such a model. The running time analysis shows that only 3.5 seconds are needed to process per entry of review, with a total token length of 407 per entry. Meanwhile, the monetary cost is approximately $1.9 per 1,000 reviews. The cost analyses further suggest that applying LLMs with our architecture is promising.

**Potential applications of identifying healthcare needs.** In this study, we propose ABSA templates used with ChatGPT to identify patients' healthcare needs and demonstrate the reliability of the results by performing well in several performance assessment scenarios. The percentage of negative reviews of the three medical categories in the ABSA template demonstrates, to some extent, the degree of patient dissatisfaction with a particular category. In other words, the higher the percentage of negative reviews, the higher the level of patient dissatisfaction in this category, and the greater the patient demands for healthcare. The model and methodology proposed in this study have a wide range of potential applications and can be used by different healthcare organizations to explore healthcare needs in depth according to different categorizations.

Leveraging big data to categorize healthcare information can reveal patterns and insights that are typically elusive. In addition, such big data can be considered a collective piece of opinions from patients' perspectives. As such, classifying such data aids governments or organizations in macro-assessing healthcare resource distribution. For instance, we can compare patient review sentiment before and after policy implementation to gauge improvements. ABSA templates can offer a nuanced categorization of potential enhancements, fostering more precise policy-making and preventing resource wastage. Governments can also use our approach to assess how healthcare resources are allocated to specific hospitals or departments, ensuring rational distribution.

In order to demonstrate the feasibility of its potential applications and its clinical value, we will use the perspective of hospital administrators as an example to show how their management analyses and allocates healthcare resources. Through regular feedback collection processes in hospitals, reviews can be obtained and the percentages of negative reviews in the three ABSA categories proposed in this paper can be computed. Categories with high negativity indicate that patients are dissatisfied with the healthcare resources in certain perspectives. With this information, the perceived performance of healthcare services can be seen at a macro level, and adjustments and improvements can be made to specific pain points if necessary. The results of our method can also be used with other downstream techniques, such as word frequency analysis and word cloud mapping [7,43], to discover the actual reasons like long waiting queues and bad attitudes of clinicians behind the negativities.

Moreover, our work can be used to compare negative percentages by region, time period, and hospital levels for gaining actionable insights into improving healthcare delivery. For example, one can compare the ABSA results of the regions with more adequate healthcare resources (e.g., Beijing in China) with the counterparts with less resources (e.g., Ningxia). We can also compare the negative percentages between different time periods, such as pre- and post-COVID, to find out the different review patterns in different healthcare dimensions. Finally, we can compare the results of different levels of hospitals (i.e., primary, secondary, tertiary) to determine the different causes of patient satisfaction and dissatisfaction.

## Comparison with existing literature

We note that there is similar work to ours. Despite this, our approach varies in some ways. The study [44] utilized LLM to select two labels for categorizing their dissatisfaction in negative patients comments, based on the ten categories set up by the authors. However, they have not optimized and demonstrated in detail the method of the use of ChatGPT, nor have they demonstrated the accuracy of their manual evaluation. Additionally, although the article

defined negative sentiment templates, but several common dissatisfaction factors were not included, such as expensive costs, lack of basic healthcare facilities and difficulty in parking.

Others surveyed the population's attitudes towards COVID-19 and used ChatGPT to generate a sentiment polarity score [45]. While their results demonstrated the power of ChatGPT in sentiment analysis, however, the scores alone can only determine people's polarities toward the disease, and cannot further analyze the reasons for dissatisfaction. In contrast, our work completely evaluated the accuracy of ChatGPT with fine-grained rationale behind dissatisfaction. In addition, the study [45] pointed out that subtle sarcasm and specialized health terminologies may affect the judgment of sentiment analysis, but we found that such problems can be alleviated after adding CoT and ABSA templates with a better accuracy.

On the other hand, a study pointed out the value of using ABSA to analyze patient comments for healthcare professionals, which can help improve the quality of healthcare [46]. However, this study also showed that the annotation data of ABSA in the healthcare domain is very sparse, which makes model training difficult [46]. Additionally, the studies about with ABSA are mainly focused on traditional deep learning and machine learning models, and no researchers have used LLM on ABSA at the time of writing [47]. Therefore, the contribution and significance of this study is to evaluate whether LLM without training can accomplish the application of ABSA in patient reviews. It can also provide LLM solution ideas for complex NLP healthcare tasks in which models are difficult to be trained, remind them to experiment with LLM on "old tasks" that were previously complex and difficult.

In sum, using LLMs in analyzing patient-generated content is a new direction and there is little in current literature. Our research opens a new direction to guide LLMs with theoretical frameworks with prompt engineering and CoT.

## Limitations

This study, while comprehensive in many respects, acknowledges certain limitations that warrant mention. Firstly, categorizing comments into three distinct categories, although beneficial for focused analysis, may still not capture the full spectrum of insights available within the data. A more granular categorization scheme could potentially unveil additional layers of information, offering a richer understanding of the sentiments expressed. Secondly, although the haodf.com platform claims that it does not censor the content of patient reviews, it is impossible to rule out the possibility of removing or suppressing negative reviews or complaints against hospitals, which could lead to a lower representation of our negative reviews, thus affecting the performance and accuracy.

Secondly, the methodologies employed did not incorporate the latest advancements in CoT, such as the Tree of Thought (ToT) or Graph of Thought (GoT) [48,49]. These cutting-edge approaches have the potential to enhance the accuracy of sentiment analysis further. Their exclusion from the current study means that there may be room for further improvement in the precision of our sentiment classification, which future research could aim to address.

On the other hand, the hyperparameters of LLM are also one of the important factors affecting the accuracy of the model [50], therefore how to optimize the hyperparameters is a meaningful research direction in the future. On the one hand, there has been a study proving the possibility of optimizing hyperparameters to improve the performance of LLM [51], methods including grid search, random search, Bayesian optimization, and evolutionary algorithms, highlighting [51]. Another study has successfully optimized hyperparameters using these methods in specific tasks [52]. However, these methods of optimizing hyperparameters are not universal and each researcher needs to restart their research on their particular task

using the above methods. Therefore, in the future, our research includes whether we can let LLM find the optimal hyperparameters by recognizing the properties of different tasks.

Finally, manual labeling of ABSA datasets is often a time-consuming and difficult process for specific ABSA tasks [47,53]. Although a test set of 2,000 entries (manually labeled) is already relatively large for a similar task, but it is still a small percentage compared to the total dataset. Meanwhile, with the development of the digital age nowadays, the dataset to be studied becomes more and more large and complex. Therefore it is not scientific to label the dataset manually. In the future, we will explore how to use LLM to automatically label the dataset, and through some methods of secondary detection of its labeled content to ensure the accuracy of the labeling. Automatic labeling technology can greatly solve the problem of scarce labeled datasets in the era of big data [54].

## Conclusions

In order to address the problem of assessing the allocation of healthcare resources from the patient's perspective, we proposed ABSA template for identifying the categories of concerns, embedded it in ChatGPT-4o, and tested its performance in several ways. The results were satisfactory and showed the feasibility of using LLMs for identifying healthcare needs from patient experience reviews. Finally, we illustrate potential applications, such as the evaluation of the effectiveness of policies on a macro level and healthcare resource allocations.

## Author contributions

**Conceptualization:** Jiaxuan Li.

**Data curation:** Jiaxuan Li, Chi Kin Lam.

**Formal analysis:** Patrick Cheong-Iao Pang.

**Funding acquisition:** Patrick Cheong-Iao Pang.

**Investigation:** Rong Chen, Chi Kin Lam.

**Methodology:** Jiaxuan Li, Yunchu Yang, Rong Chen, Dennis Wong.

**Project administration:** Yunchu Yang, Dashun Zheng, Dennis Wong.

**Resources:** Jiaxuan Li, Patrick Cheong-Iao Pang.

**Software:** Yunchu Yang, Dashun Zheng, Dennis Wong.

**Supervision:** Patrick Cheong-Iao Pang, Dennis Wong, Yapeng Wang.

**Validation:** Yunchu Yang, Rong Chen, Yapeng Wang.

**Visualization:** Rong Chen, Dashun Zheng, Chi Kin Lam.

**Writing – original draft:** Jiaxuan Li.

**Writing – review & editing:** Patrick Cheong-Iao Pang, Yapeng Wang.

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
