## [Decision Letter · Decision Letter 0]

4 Sep 2024

PONE-D-24-34064Identifying Healthcare Needs with Patient Experience Reviews Using ChatGPTPLOS ONE

Dear Dr. Cheong-lao Pang,

Thank you for submitting your manuscript to PLOS ONE. After careful consideration, we feel that it has merit but does not fully meet PLOS ONE’s publication criteria as it currently stands. Therefore, we invite you to submit a revised version of the manuscript that addresses the points raised during the review process.

We look forward to receiving your revised manuscript.

Kind regards,

Jinran Wu, PhD

Academic Editor

PLOS ONE

**Journal requirements: **

This research was supported by the Macao Science and Technology Development Fund (funding IDs: 0048/2021/APD; 0088/2023/ITP2).

4. We note that you have referenced "Mahapatra P." which has currently not yet been accepted for publication. Please remove this from your References and amend this to state in the body of your manuscript: (Mahapatra P. [Unpublished]”) as detailed online in our guide for authors

Reviewers' comments:

Reviewer's Responses to Questions

**Comments to the Author**

1. Is the manuscript technically sound, and do the data support the conclusions?

Reviewer #1: Partly

Reviewer #2: Yes

Reviewer #3: Yes

Reviewer #4: Yes

2. Has the statistical analysis been performed appropriately and rigorously? 

Reviewer #1: Yes

Reviewer #2: Yes

Reviewer #3: Yes

Reviewer #4: Yes

3. Have the authors made all data underlying the findings in their manuscript fully available?

Reviewer #1: Yes

Reviewer #2: Yes

Reviewer #3: Yes

Reviewer #4: No

4. Is the manuscript presented in an intelligible fashion and written in standard English?

Reviewer #1: Yes

Reviewer #2: Yes

Reviewer #3: Yes

Reviewer #4: Yes

5. Review Comments to the Author

**Reviewer #1:**  This work used the reviews to create Aspect-Based Sentiment Analysis (ABSA) templates, categorizing patient reviews into three categories that reflect patients' areas of concern. By introducing thought chains, the ABSA templates were embedded into prompts for ChatGPT, which was then used to identify patient needs. The results show that the proposed method achieved a weighted total precision of 0.907, which is outstanding compared to the direct narrative tasks in ChatGPT.

Overall, this work is complete, but there is still room for improvement. My suggestions are as follows:

1.Lines 53-54 “but these techniques still struggle to pinpoint specific areas of patient dissatisfaction within individual reviews” : is there any literature to support this statement?

2.Lines 51 - 59：In this paragraph, the first part ("Recent research... individual reviews") discusses the limitations of deep learning and data mining approaches in analyzing patient reviews, but the second part ("Finally, some studies... patient experience") covers other topics. These two parts lack a clear logical connection. It is recommended to revise this paragraph to ensure clearer and more coherent logic.

3.Lines 112 -123 : The hyperparameters within ChatGPT are set to their default values. Is it possible to ensure that the parameters used are optimal without tuning them? Could the model's performance be improved by adjusting these hyperparameters?

4.The test size is 2000, out of a total data size of 504,198. This is a relatively small proportion (approximately 0.4%) for the test set, which might not provide a robust estimate of the model's performance on unseen data.

5.Lines 245- 246: For the test set (2000 samples), labeling was done manually. For larger datasets, relying on manual labeling is very inefficient. Are there any better methods to improve efficiency?

6.It is recommended to add an explanation of how the three metrics (Precision, Recall, and F1-weight) are used to evaluate the model's performance. For example, the higher the Precision, the better the model's performance.

7.Does the F1 score in Table 3 represent the weighted F1 score ? And is the F1 score in the Model Performance section also referring to the weighted F1 score? The authors should provide a clear description.

**Reviewer #2: ** The authors need more literatures to support their claim as in the paper there is a need to justify evidences with literatures that can prove most of the claim are true and can ensure that this is a worth area to research on.

**Reviewer #3:**  here are several review comments for the manuscript:

1 The manuscript is well-organized, and the objectives are clearly stated. However, the authors could enhance the clarity by providing a more detailed introduction to the significance of the study and its contribution to the existing literature.

2 the authors should elaborate on the methodology, particularly the process of creating ABSA templates and the rationale behind the chosen categories. Additionally, a discussion on the potential limitations of ChatGPT in understanding complex sentiment expressions would be beneficial.

3 Comparison with Existing Studies: The authors should provide a more comprehensive comparison with other studies that have used similar approaches or have addressed the same research questions. This will help position their work within the broader context of the field.

4 It is recommended that a professional editor review the manuscript for language improvements.

**Reviewer #4: ** This study explores the use of ChatGPT combined with Aspect-Based Sentiment Analysis (ABSA) templates to identify healthcare needs from patient experience reviews. The researchers analyzed 504,198 reviews from an online medical platform, categorizing patient feedback into three areas: patient experience, physician skills and efficiency, and infrastructure and administration. By embedding ABSA templates into ChatGPT prompts, the study achieved higher precision, recall, and F1 scores compared to direct narrative tasks in ChatGPT. The results suggest that this approach can effectively capture patient sentiments and provide valuable insights for healthcare resource allocation and policy evaluation. The study concludes that integrating LLMs with structured analysis techniques like ABSA can significantly enhance the understanding of patient needs in healthcare.

I have attached a few comments, with the main concern being the use of an older version of ChatGPT, which has been documented to have lower performance both as a general LLM and specifically in healthcare-related studies. Given that the aim is to optimize performance while saving costs, and considering that time efficiency is also a factor in cost, I strongly recommend using the most advanced version available.

Abstract:

The first sentence of the objective section should be moved to the introduction.

Introduction:

1 - In lines 55 and 58, please address the redundancy of the phrase "It has also been pointed out," which appears twice in close proximity.

2 - The last two paragraphs of the introduction currently summarize the paper and should be moved to the discussion section instead. Additionally, the introduction is quite lengthy. I suggest removing these parts and making the introduction more concise.

3 Methods – line 235: It's unfortunate that significant effort was invested while using ChatGPT-3.5, especially since numerous studies have already demonstrated that newer versions are significantly more accurate. Given the goal of "avoiding unnecessary waste of healthcare resources," as stated in the abstract, I would strongly recommend using the latest paid version, which is relatively affordable and would likely offer more cost-effective and precise results.

Discussion:

4 - Innovative Aspect: The integration of ABSA templates with ChatGPT is well-executed, but the authors should more clearly emphasize what sets their approach apart from existing methods. approach innovative compared to existing methods. Highlighting the unique contributions more explicitly would make the manuscript's novelty clearer.

6. PLOS authors have the option to publish the peer review history of their article (what does this mean? ). If published, this will include your full peer review and any attached files.

**Do you want your identity to be public for this peer review?** For information about this choice, including consent withdrawal, please see our Privacy Policy .

Reviewer #1: No

Reviewer #2: **Yes: ** Abdullahi Tunde Aborode

Reviewer #3: No

Reviewer #4: No

---

## [Author Response · Author response to Decision Letter 1]

17 Oct 2024

Dear Editor and Reviewers,

Many thanks for your comments on our manuscript and providing us the opportunity of submitting a revised version for your consideration. We have addressed all the comments and made significant changes to the manuscript accordingly. The changes to the manuscript are red highlighted. Specific responses to each comment are detailed below.

Reviewer 1#:

General comments

This work used the reviews to create Aspect-Based Sentiment Analysis (ABSA) templates, categorizing patient reviews into three categories that reflect patients' areas of concern. By introducing thought chains, the ABSA templates were embedded into prompts for ChatGPT, which was then used to identify patient needs. The results show that the proposed method achieved a weighted total precision of 0.907, which is outstanding compared to the direct narrative tasks in ChatGPT. Overall, this work is complete, but there is still room for improvement. My suggestions are as follows:

RESPONSE:

We appreciate the reviewers' valuable suggestions and recognition of our work. Through which we have improved the quality of the paper, and hope that our revisions will address these issues.

Specific comments

1. Lines 53-54 “but these techniques still struggle to pinpoint specific areas of patient dissatisfaction within individual reviews”: is there any literature to support this statement?

RESPONSE: We thank the reviewers for the suggestion that our previous lack of cited literature failed to support this sentence. In the latest version, we have cited two recent studies (references 43-44) from the last two years to support the sentence.

2. Lines 51 - 59: In this paragraph, the first part ("Recent research... individual reviews") discusses the limitations of deep learning and data mining approaches in analyzing patient reviews, but the second part ("Finally, some studies... patient experience") covers other topics. These two parts lack a clear logical connection. It is recommended to revise this paragraph to ensure clearer and more coherent logic.

RESPONSE: Thanks to the reviewers' meaningful comments. We were missing a logical connection to the patient experience in this section. In the new version, we have added an introduction about the importance of patient experience and some connecting statements to make the context more logical.

3. Lines 112 -123 : The hyperparameters within ChatGPT are set to their default values. Is it possible to ensure that the parameters used are optimal without tuning them? Could the model's performance be improved by adjusting these hyperparameters?

RESPONSE: Thanks to the reviewer for the comments. On the one hand, we wanted to see if LLM could accomplish this complex task without adjusting hyperparameters (the easiest to use version). On the other hand, the hyperparameters research on LLM is a hot and difficult research direction at present, especially the hyperparameters research on closed-source LLM. But we lacked a description of this in the paper, so we include this aspect in Limitations and discuss the possibility of a more detailed study in the future, including how to find the optimal hyperparameters for a specific task.

4. The test size is 2000, out of a total data size of 504,198. This is a relatively small proportion (approximately 0.4%) for the test set, which might not provide a robust estimate of the model's performance on unseen data.

RESPONSE: Thanks to the reviewer's comments. We noticed this issue during our experiments, so we included Reliability and Stability Evaluation (Lines 282) for a secondary assessment of accuracy. The reason for the sparse test set is that the labeling process in the ABSA task is very difficult [1-3], and we already have one of the largest test set data in the same type of experiment (manual labeling). Therefore, in the latest version, we have included a description of this aspect in the Limitation section, as well as a discussion of the fact that in the future we can do related research on automatic labeling of LLMs to address the difficulty of labeling in complex tasks.

[1]Hua Y C, Denny P, Wicker J, et al. A systematic review of aspect-based sentiment analysis: domains, methods, and trends. Artificial Intelligence Review, 2024, 57(11): 296.

[2]Imani M, Noferesti S. Aspect extraction and classification for sentiment analysis in drug reviews. Journal of Intelligent Information Systems, 2022, 59(3): 613-633.

[3]Chen Y, Liu S, Zhang X, et al. Automatically labeled data generation for large scale event extraction. Proceedings of the 55th Annual Meeting of the Association for Computational Linguistics (Volume 1: Long Papers). 2017: 409-419.

5. Lines 245- 246: For the test set (2000 samples), labeling was done manually. For larger datasets, relying on manual labeling is very inefficient. Are there any better methods to improve efficiency?

RESPONSE: Thanks to the reviewers for their suggestions. And as mentioned in the previous response, improving labeling efficiency is a pressing issue. Therefore, we discuss in Limitation about how to use LLM automatically labeled datasets in the future to improve labeling efficiency. We hope this will address your query.

6. It is recommended to add an explanation of how the three metrics (Precision, Recall, and F1-weight) are used to evaluate the model's performance. For example, the higher the Precision, the better the model's performance.

RESPONSE: Thanks to the reviewer's suggestion. We omitted this explanation before. In the new version, we have included an explanation of how the three metrics specific to this task relate to model performance in line 293.

7. Does the F1 score in Table 3 represent the weighted F1 score ? And is the F1 score in the Model Performance section also referring to the weighted F1 score? The authors should provide a clear description.

RESPONSE: Thanks to the reviewer for such a detailed comment. We have used a weighted F1 score in order to avoid sample imbalance, and have abbreviated it to an F1 score for the sake of the aesthetics of the table as well as the readability of the paper. But we forgot to introduce it, in the latest version, we have added this note and informed that the F1 score in the entire Result section is a weighted F1 score.

Reviewer 2#:

General comments

The authors need more literatures to support their claim as in the paper there is a need to justify evidences with literatures that can prove most of the claim are true and can ensure that this is a worth area to research on.

RESPONSE: We thank the reviewers for their valuable suggestions. And we have made a complete revision to address this issue. In this latest version, we have included new literature in several sections of the text to support our conclusions and elaborations. The total number of new papers added is more than ten, and we hope that the reviewers' questions can be solved.

Reviewer 3#:

General comments

Here are several review comments for the manuscript:

RESPONSE: We thank the reviewers for their valuable comments. This comments greatly improved the quality of our paper.

Specific comments

1. The manuscript is well-organized, and the objectives are clearly stated. However, the authors could enhance the clarity by providing a more detailed introduction to the significance of the study and its contribution to the existing literature.

RESPONSE: Thanks to the reviewers' valuable comments. We have presented the significance of the study in more detail in the latest version, which mainly includes the solution to the difficulties of ABSA in training models and the solution ideas in complex tasks, etc. On the other hand, we also elaborates our value and contribution to the existing literature.

2. The authors should elaborate on the methodology, particularly the process of creating ABSA templates and the rationale behind the chosen categories. Additionally, a discussion on the potential limitations of ChatGPT in understanding complex sentiment expressions would be beneficial.

RESPONSE: Thanks to the reviewers' valuable comments. We have elaborated the process of creating ABSA in more detail in our latest version and explained the rationale behind the selected categories with specific examples and the particularities of the Chinese healthcare environment. Finally, we have included in the Model Accuracy and Feasibility section of the Discussion section we describe the limitations of LLM in handling complex tasks and illustrate them using concrete examples.

3.Comparison with Existing Studies: The authors should provide a more comprehensive comparison with other studies that have used similar approaches or have addressed the same research questions. This will help position their work within the broader context of the field.

RESPONSE: Thanks to the reviewers for their meaningful comments. We have included some new studies for comparison in our latest version to carry out a more in-depth discussion on the significance and contribution of our approach. Specifically, we have cited one study that addresses the same research question as ours and described their research method shortcomings. On the other hand, we cite a review article published in 2024 on the ABSA methodology, and in this way we compare all the studies using similar methodologies and describe the research gaps in the ABSA field on LLM.

4.It is recommended that a professional editor review the manuscript for language improvements.

RESPONSE: Thanks to the reviewer's suggestion. We have addressed the language issues after fully reviewing our manuscript.

Reviewer 4#:

General comments

This study explores the use of ChatGPT combined with Aspect-Based Sentiment Analysis (ABSA) templates to identify healthcare needs from patient experience reviews. The researchers analyzed 504,198 reviews from an online medical platform, categorizing patient feedback into three areas: patient experience, physician skills and efficiency, and infrastructure and administration. By embedding ABSA templates into ChatGPT prompts, the study achieved higher precision, recall, and F1 scores compared to direct narrative tasks in ChatGPT. The results suggest that this approach can effectively capture patient sentiments and provide valuable insights for healthcare resource allocation and policy evaluation. The study concludes that integrating LLMs with structured analysis techniques like ABSA can significantly enhance the understanding of patient needs in healthcare. I have attached a few comments, with the main concern being the use of an older version of ChatGPT, which has been documented to have lower performance both as a general LLM and specifically in healthcare-related studies. Given that the aim is to optimize performance while saving costs, and considering that time efficiency is also a factor in cost, I strongly recommend using the most advanced version available.

RESPONSE: We thank the reviewers for such detailed and sincere comments. We re-experimented with the latest ChatGPT-4o-lastest at the first opportunity after receipt of the comments and reviewed the accuracy using the same multiplicity of methods. We changed all the descriptions of the older versions of ChatGPT in the article as well as changed the previous F1 score, accuracy and so on.

Specific comments

1. Abstract: The first sentence of the objective section should be moved to the introduction.

RESPONSE: Thanks to the reviewer's comments. We have moved this sentence to the Introduction section as you suggested and added two new citations to improve the reliability of this sentence.

2.Introduction: In lines 55 and 58, please address the redundancy of the phrase "It has also been pointed out," which appears twice in close proximity.

RESPONSE: Thanks to the reviewers' valuable suggestions. We have addressed this writing error in the latest version, and we have checked the full text to ensure that similar errors do not occur again.

3. Introduction: The last two paragraphs of the introduction currently summarize the paper and should be moved to the discussion section instead. Additionally, the introduction is quite lengthy. I suggest removing these parts and making the introduction more concise.

RESPONSE: Thanks to the reviewer's meaningful comments. Our introductory section was indeed too long resulting in a loss of readability, so we removed the paragraph to summarize the paper as per the reviewer's suggestion as a way to ensure brevity in the introductory section.

4. Methods: line 235: It's unfortunate that significant effort was invested while using ChatGPT-3.5, especially since numerous studies have already demonstrated that newer versions are significantly more accurate. Given the goal of "avoiding unnecessary waste of healthcare resources," as stated in the abstract, I would strongly recommend using the latest paid version, which is relatively affordable and would likely offer more cost-effective and precise results.

RESPONSE: Thank you for such a detailed suggestion, and we agreed made a lot of sense. We redid the entire experiment with ChatGPT-4o. As you said, benefiting from the advantages of the new model in terms of cost and efficiency, we successfully completed the experiment in a short time and covered it with the original model in the full text.

5. Discussion: Innovative Aspect: The integration of ABSA templates with ChatGPT is well-executed, but the authors should more clearly emphasize what sets their approach apart from existing methods. approach innovative compared to existing methods. Highlighting the unique contributions more explicitly would make the manuscript's novelty clearer.

RESPONSE: Thanks to the reviewers' valuable comments. In this latest version we cite a review article published in 2024 on ABSA methods, comparing the differences between all of them and describing issues such as the use of LLM in ABSA being a research gap. On the other hand, we cite the ABSA study, also on patient reviews, and describe how the challenge of this similar task lies in model training, among other things. Finally, we present the unique contributions and implications of our study through these new citations.

---

## [Decision Letter · Decision Letter 1]

24 Oct 2024

Identifying Healthcare Needs with Patient Experience Reviews Using ChatGPT

PONE-D-24-34064R1

Dear Dr. Pang,

We’re pleased to inform you that your manuscript has been judged scientifically suitable for publication and will be formally accepted for publication once it meets all outstanding technical requirements.

Kind regards,

Jinran Wu, PhD

Academic Editor

PLOS ONE

Reviewers' comments:

Reviewer's Responses to Questions

**Comments to the Author**

1. If the authors have adequately addressed your comments raised in a previous round of review and you feel that this manuscript is now acceptable for publication, you may indicate that here to bypass the “Comments to the Author” section, enter your conflict of interest statement in the “Confidential to Editor” section, and submit your "Accept" recommendation.

Reviewer #1: All comments have been addressed

Reviewer #2: All comments have been addressed

2. Is the manuscript technically sound, and do the data support the conclusions?

Reviewer #1: Yes

Reviewer #2: Yes

3. Has the statistical analysis been performed appropriately and rigorously? 

Reviewer #1: Yes

Reviewer #2: No

4. Have the authors made all data underlying the findings in their manuscript fully available?

Reviewer #1: Yes

Reviewer #2: Yes

5. Is the manuscript presented in an intelligible fashion and written in standard English?

Reviewer #1: Yes

Reviewer #2: Yes

6. Review Comments to the Author

Reviewer #1: (No Response)

Reviewer #2: The authors have addressed the comments as suggested with detailed feedback which have make the paper worthy.

---

## [Editor Report · Acceptance letter]

PONE-D-24-34064R1

PLOS ONE

Dear Dr. Pang,

I'm pleased to inform you that your manuscript has been deemed suitable for publication in PLOS ONE. Congratulations! Your manuscript is now being handed over to our production team.

Kind regards,

on behalf of

Dr. Jinran Wu

Academic Editor

PLOS ONE